# Implication of Vaginal and Cesarean Section Delivery Method in Black–White Differentials in Infant Mortality in the United States: Linked Birth/Infant Death Records, 2007–2016

**DOI:** 10.3390/ijerph17093146

**Published:** 2020-04-30

**Authors:** Laurens Holmes Jr., Leah O’Neill, Hikma Elmi, Chinaka Chinacherem, Camillia Comeaux, Lavisha Pelaez, Kirk W. Dabney, Olumuyiwa Akinola, Michael Enwere

**Affiliations:** 1Nemours Children’s Health System, Wilmington, DE 19802, USA; leah.oneill.16@cnu.edu (L.O.); Hikma.Elmi@nemours.org (H.E.); Chinaka.Chinacherem@nemours.org (C.C.); Camillia.Comeaux@nemours.org (C.C.); lavisha.pelaez@nemours.org (L.P.); kirk.dabney@nemours.org (K.W.D.); Tolu.Akinola@nemours.org (O.A.); mikky89@gmail.com (M.E.); 2Department of Biological Sciences, University of Delaware, Newark, DE 19716, USA; 3Department of Molecular Biology & Chemistry, Christopher Newport University, Newport News, VA 23606, USA; 4College of Pharmaceutical Science, Institute of Public Health, Florida A & M University, Tallahassee, FL 32307, USA; 5Department of Public Health, Walden University, Minneapolis, MN 55401, USA

**Keywords:** vaginal, cesarean section, African American women, infant mortality, race/ethnicity

## Abstract

Racial/ethnic disparities in infant mortality (IM) continue to persist in the United States, with Black/African Americans (AA) being disproportionally affected with a three-fold increase in mortality compared to Whites. Epidemiological data have identified maternal characteristics in IM risk such as preeclampsia, eclampsia, maternal education, smoking, maternal weight, maternal socioeconomic status (SES), and family structure. Understanding the social gradient in health including implicit bias, as inherent in the method of labor and delivery and the racial heterogeneity, may facilitate intervention mapping in narrowing the Black–White IM risk differences. We aimed to assess the temporal/racial trends and the methods of delivery, mainly vaginal vs. cesarean section (C-section) as an exposure function of IM. The United States linked birth/infant death records (2007–2016) were used with a cross-sectional ecological design. The analysis involved chi squared statistic, incidence rate estimation by binomial regression model, and period percent change. Of the 40,445,070 births between 2007 and 2016, cumulative mortality incidence was 249,135 (1.16 per 1000). The IM rate was highest among Black/AA (11.41 per 1000), intermediate among Whites (5.19 per 1000), and lowest among Asian /Pacific Islanders (4.24 per 1000). The cumulative incidence rate difference, comparing vaginal to cesarean procedure was 1.73 per 1000 infants, implying excess IM with C-section. Compared to C-section, there was a 31% decreased risk of IM among mothers with vaginal delivery, rate ratio (RR) = 0.69, 95% confidence interval (CI): 0.64–0.74. Racial disparities were observed in the method of delivery associated with IM. Black/AA mothers with vaginal delivery had a 6% decreased risk of IM compared to C-section, RR = 0.94, 95% CI: 0.92–0.95, while Whites with vaginal delivery had a 38% decrease risk of IM relative to C-section, RR= 0.68, 95% CI: 0.67–0.69, *p* < 0.001. Infant mortality varied by race, with Black/AA disproportionally affected, which is explained in part by labor and delivery procedures, suggestive of reliable and equitable intrapartum assessment of Black/AA mothers during labor, as well as implicit bias marginalization in the healthcare system.

## 1. Introduction

Since the creation of the National Center for Health Statistics (NCHS), the United States (U.S.) has experienced racial/ethnic differences in morbidity and mortality, where Black/African American (AA) children experienced excess rates of chronic diseases and untimely deaths compared to their White counterparts, which is explained in part by built environment, institutional, or structural racism, crime, and segregation [1]. Structural racism reflects the unfair treatment of Black/AA at employment, where blacks are less likely to be employed in high paying jobs (indicative of low socioeconomic status) and decreased access to healthcare via health insurance. Likewise, absence of exclusive breastfeeding during the first six months of life, driven by low wage employment of Black mothers, contributes to poor health outcomes of Black children, resulting in infant mortality. The National Institutes of Health (NIH), has indicted that the inability for United States to achieve its full health potentials is mainly a result of the health disparities, which reflects subpopulation differences in social, economic, and environmental conditions related to health [2]. These health disparities are also demonstrated in African American women who are twice as likely to experience life-threatening pregnancy-related complications, also known as severe maternal morbidity (SMM) [3,4]. Infant mortality, which includes neonatal deaths, reflects deaths before the first birthday. Because pregnancy-related complications are correlated with infant mortality, African Americans (AA)/Black mothers are more than twice as likely to have children who experience mortality within the first 12 months of birth. Previous studies have observed the correlation between low socioeconomic status and excess morbidity and mortality in the U.S. population, which explains in part the excess infant mortality among Black/AA [5]. The U.S. racial differences in infant mortality continue to persist, with Blacks/AA children being 2–3 times as likely to die compared to Whites. The observed disparities may be explained by racial variances in maternal factors, mainly pre-term birth, low birth weight, obesity, diabetes, hypertension, pre-eclampsia, alcohol, tobacco, HIV, rural residence, race, prenatal visits and psychosocial stressors. A study on preterm weight observed increased likelihood of infant mortality with low birth weight and very small for gestational age (VSGA). Children with VSGA (<2nd percentile vs. >50th–74th percentile) had increased risk of infant mortality in a large cohort (*n* = 4525) of infants without congenital anomalies [6].

In the U.S., the Center for Disease Control and Prevention (CDC) provides a comprehensive listing of infant mortality, by clinical conditions and by mother’s self-identified race and/or ethnicity, from 1968 to 2015. The predisposing factors or causes of infant mortality remain to be fully understood despite the race differences in risk by clinical disorders. With the multifactorial etiology of infant mortality and stillbirth, the cause of causes (implying the etiopathogenesis of the racial differences such, as social gradient) remain unclear and not very well understood. However, an investigation on the clinical conditions, such as the method of labor and delivery namely cesarean section (C-section) and vaginal deliveries, resulting in the observed differences in infant mortality remains a viable and pragmatic pathway of an explanatory epidemiologic model.

Cesarean and vaginal delivery are two forms of delivery methods. Even though vaginal deliveries occur twice as often as C-section deliveries, vaginal deliveries are declining while C-section deliveries are increasing. More specifically, there are two different types of vaginal deliveries: vacuum assisted and forceps assisted. From 2005 to 2013, rates of both vacuum assisted (from 5.8% to 4.1%) and forceps assisted (from 1.4% to 0.9%) vaginal deliveries decreased [7]. In contrast, the number of cesarean deliveries has increased from 31.9% in 2016 to 32% in 2017 [8]. Even though this difference is marginal, it represents increasing trends in elective cesarean deliveries [9]. With such a higher rate of cesarean deliveries, there is an increased risk of maternal and fetal complications, such as maternal death, infant mortality, hemorrhage, infection, incidental surgical injuries, and extended hospitalization. The rates of these complications are higher in incidences of cesarean deliveries than vaginal deliveries [10]. Studies have shown that cesarean delivery rates are positively correlated with infant mortality rates among industrialized countries [11]. Available epidemiological data have indicated a disparity in cesarean delivery rates and infant mortality rates between non-Hispanic Black and White women. Non-Hispanic Black women have a higher rate of cesarean delivery (36%) compared to non-Hispanic White women (30.9%) [12]. With the aggregate data on the methods of delivery, the current study aimed to examine the racial differences in infant mortality comparing cesarean with vaginal delivery. Additionally, we sought to utilize maternal education in explaining the racial variance in the method of delivery as the exposure function of infant mortality.

## 2. Methods

This study was conducted to examine the exposure function of the method of labor and delivery in racial differentials in infant mortality. We aimed to assess the implication of method of labor and delivery, child factors (e.g., birth weight), and maternal education in infant mortality variances by race/ethnicity, using data from the Center of Disease Control and Prevention (CDC). The availability of these data allows for the assessment of specific subpopulations, given the multifactorial etiology in infant mortality. Therefore, the understanding of factors, especially healthcare delivery system, contributory effect (labor and delivery method) associated with the excess infant deaths among Black/AA will facilitate intervention mapping and subsequent risk reduction; thus, transforming health equity in this dimension. After an institutional review board (IRB) approval from the Nemours Healthcare System for Children, we conducted a study to assess the relationship between infant mortality and method of delivery, and racial heterogeneity therein.

Data from the National Center for Health Statistics (NCHS, 2007–2016) were used to examine infant death prevalence by race/ethnicity, as well as to determine the labor and delivery method associated with Black–White differences in infant mortality. This data set represent the United States linked birth/infant death records, 2007–2016. As an aggregate database, the Center for Disease Control provides information on death records by year of death of the infant, the race of the infant as mother’s bridge race, maternal education, sex of the child, weight of the child, attending healthcare provider, and geography (location/region, urbanization, and state). The details of the utilized database are available elsewhere (www.wonder.cdc.gov).

### 2.1. Study Design

A cross-sectional ecological non-experimental design was used to examine Black–White risk differences in infant mortality, with the method of delivery as the exposure function of the variance. This design involves the utilization of pre-existing aggregate data, which is methodologically feasible, given the simultaneous gathering of the variables in the dataset. Additionally, this design allows for the examination of other independent variables as potential exposures or confounding variables.

### 2.2. Study Eligibility

We used a consecutive sample which represents a probability sample given its sampling representativeness (*n* = 40,445,070). To estimate the power, implying the ability of the study to detect a clinically meaningful differences between race in infant mortality, we estimated the power using the following parameters, sample size of American Indian/Alaska Native (AI/AN), *n* = 469,563, which was the smallest sample size in the subpopulation by race. The effect size (Δ) = 0.20 (20%). The type I error tolerance was 0.05 (5%), implying a 95% confidence interval (CI). With these parameters, we estimated the statistical power (type II error tolerance <20%) as a power sufficient to detect a minimum difference of 20% in comparing the mortality experience of Black/AA relative to Whites, with respect to the delivery method, should such a difference exist.

### 2.3. Statistical Analysis

A pre-analysis screening tool was used to examine the dataset for accuracy in terms of data gathering based on the linked birth/infant death records from 2007–2016. The descriptive statistics were performed to examine the distribution of the study variables by the main independent variable (race) and methods of delivery frequencies and percentages, while annual and period percent change was used to examine the temporal trends by rates from 2007–2017. The chi square statistic, incidence rate ratio, and tabulation analysis were used to assess the variables’ independence as well as the association between independent (race and delivery method) and outcome (infant mortality) variables. The cumulative incidence risk was assessed using binomial regression model, implying the risk ratio, comparing categories or domain, such as race with one subcategory as the referent or reference group in the risk estimation. This approach was statistically accurate and feasible, as there was no assumption of normality for the outcome or response variables in the application of binomial statistical modeling in evidence discovery. The period prevalence was assessed using a standardized formula, which determined whether or not method of delivery, with respect to mortality, remained stable or changed over a specified period of time:
Period percent change = current or final period or year (rate—initial or previous period or year (rate)/initial or previous period or year (rate) × 100.

The type 1 error tolerance 5% (95% CI) and all tests were two-tailed. All analyses were performed using STATA statistical software (Version 15, STATA Cooperation, College Station, TX, USA).

## 3. Results

The data in this study represent the United States linked birth/infant death records, from 2007–2016. Of the 40,445,070 births between 2007 and 2016, there were 249,135 deaths (6.16 per 1000). The cumulative infant mortality rate during this period was 6.16 per 1000 infants. This rate varied by race of the infants. The rates were highest among Black/AA (11.41 per 1000), intermediate among American Indian/Alaskan Natives (8.32 per 1000), and lowest among Whites (5.19 per 1000) and Asian/Pacific Islanders (4.24 per 1000). The rates also varied by the method of delivery, and the infant mortality related to cesarean section (8.49 per 1000) was higher compared to vaginal delivery (6.75 per 1000), implying a 1.74 per 1000 infant mortality rate difference.

Table 1 illustrates vaginal and cesarean delivery rates, comparing American Indian/Alaskan Native, Asian/Pacific Islander, Black/AA, and White mothers. There were racial differences in the infant mortality rate. The rates were highest among Black/AA (vaginal, 1.1% vs. caesarian, 1.2%), intermediate among American Indian/Alaskan Natives (vaginal, 0.7% vs. caesarian, 1.1%), and lowest among Whites (vaginal, 0.5% vs. caesarian, 0.7%) and Asian/Pacific Islanders (vaginal, 0.4% vs. caesarian, 0.5%). The rates of infant mortality by vaginal and cesarean section stratified by race. Generally, irrespective of race, vaginal delivery indicated a lower risk of infant mortality relative to C-section, implying the protective effect of vaginal delivery on overall U.S. infant mortality. Among Whites, infant mortality associated with vaginal delivery was 0.5%, while among Black/AA it was 1.1%, implying a 0.6% infant mortality risk difference when comparing Black/AA and Whites in this method of delivery. Similarly, infant mortality varied by C-section when comparing Black to White infants. Among White infants, C-section accounted for 0.7% of overall mortality, while among Black/AA was 1.2%, χ^2^ (4) = 40,000, *p* < 0.001. There was a statistically significant difference in infant mortality by the method of delivery among American Indians/Alaskan Natives. Vaginal delivery was associated with 0.7% while cesarean section was associated with 1.1% infant mortality χ^2^ (4) = 40,000, *p* < 0.001. Among American Indians/Alaskan Natives vaginal delivery was protective against infant mortality. Compared to C-section, there was a 31% decreased risk of infant mortality, incidence rate ratio = 0.69, 95% CI: 0.64–0.74, *p* < 0.001.

Table 2 demonstrates the overall incidence rate ratio in infant mortality risk stratified by race comparing vaginal and C-section births. The infant mortality risk associated with C-section was lowest among Whites, as well as American Indians/Alaskan Natives. Compared to C-sections in these two races, there were 32% and 31% decreased risk of infant mortality among Whites and American Indians/Alaskan Natives respectively, when comparing vaginal to C-section methods of delivery. In contrast, the incidence rate ratio difference was very marginal among Black/AA. Compared with C-section methods, there was 6% decreased risk of infant mortality for vaginal deliveries. A statistically significant difference in mortality was observed by the method of delivery among Asian/Pacific Islanders. The vaginal delivery method in this population was associated with 0.4% while C-section was associated with a 0.5% of infant mortality, χ^2^ (1) = 90.6, *p* < 0.001.

Among Asian/Pacific Islanders, there was a decreased risk of infant mortality following vaginal delivery relative to cesarean section. Compared with C-section, there was a 17% decreased risk of infant mortality, incidence rate ratio (IRR) = 0.83, 95% CI: 0.80–0.86, *p* < 0.001. The incidence rate of dying, given the method of delivery, indicated increased risk among Black/African American mothers who underwent cesarean section. Compared to C-section, there was a 6% decreased risk of infant mortality among mothers with vaginal delivery, IRR = 0.94, 95% CI: 0.92–0.95, *p* < 0.001. Similarly, there was a statistically significant difference in infant mortality by the method of delivery for Whites. The infant mortality was higher with C-section compared to vaginal delivery (0.45% vs. 0.66%), χ^2^ (1) = 5600, *p* < 0.001. Among White mothers, vaginal delivery was protective against infant mortality. White mothers who underwent vaginal delivery had 32% decreased risk infant mortality, IRR = 0.68, 95% CI: 0.67–0.69, *p* < 0.001.

Table 3 presents the method of delivery stratified by maternal education and race implying the effects of race and education on infant mortality in the United States. Regardless of race, infant mortality was highest among mothers with less than high school education and lowest among mothers with a post-graduate degree. Among American Indian/Alaskan Native, mothers with less than high school education, vaginal delivery was associated with 9.33 per 1000, while C-section was associated with 14.3 per 1000. Similarly, Black/African American infant death among mothers who possessed less than high school education was 9.8 per 1000 and 13.0 per 1000 for vaginal and cesarean, respectively. A similar pattern was observed among White and Asian/Pacific Islander mothers. Among White mothers with less than high school education, vaginal delivery was associated with 5.5 per 1000, while C-section was associated with 8.85 per 1000. Additionally, White mothers a with post-graduate degree experienced the lowest infant mortality associated with vaginal delivery 2.5 per 1000 during this time interval.

Table 4 describes the rate of infant mortality related to the method of delivery, race, and the sex of the infants. Regardless of the method of delivery or race, the risk of dying as an infant was higher for males compared to females. Among American Indian/Alaskan Natives, females delivered through vaginal method were 20% less likely to die relative to male, IRR = 0.80, 95% CI: 0.74–0.87. Similarly, with respect to cesarean section, females were 18% less likely to die among American Indian/Alaskan Native, IRR = 0.82, 95% CI: 0.74–0.91. Among Black/AA mothers, female infants were less likely to die regardless of the method of delivery compared to male infants. Regarding vaginal delivery Black/AA infant females were 21% less likely to die compared to males, IRR = 0.79, 95% CI: 0.78–0.81. In addition, concerning C-section there was an 11% decreased risk of infant mortality among males compared to vaginal method, IRR = 0.89, 95% CI: 0.87–0.90.

Table 5 exhibits the cumulative infant mortality rates by five-year intervals and the percent change by race and method of labor and delivery. Regardless of race, the cesarean section method of delivery was higher in the first period (2007–2011) of the five-year intervals relative to the second period (2012–2016). The highest rate in the C-section was observed among Black/AA with 12.63 per 1000, while the lowest rate was observed among Asian/Pacific Islanders with 4.61 per 1000. The infant mortality rate slightly lowered in the second period, with the lowest rate observed among Asian/Pacific Islanders despite the slight increase in this period (2007–2011) with 4.97 per 1000. The lowest infant mortality rate in the first period was observed among Asian/Pacific Islanders (3.74 per 1000), while the highest was observed among Black/AA (11.60 per 1000). In these two subpopulations, namely Asian/PI and Black/AA, rates slightly increased in the second period. With respect to percent change, a positive percent change was observed in cesarean delivery (CD) among American Indian/Alaskan Natives, while negative trends were observed in either C-section or vaginal delivery (VD) among other racial groups, indicative of infant mortality overall reduction despite the highest rates among Black/AA and lowest rates among Whites and Asian/Pacific Islander. The trends and the period percent change are indicated in Figure 1. There was overall increase in infant mortality in the first period relative to the second period. The period percent change indicated reduction in infant mortality with non-substantial effect on the temporal trends on the risk of dying, R^2^ = 2%.

## 4. Discussion

The current study assessed the exposure function of labor and delivery methods in the persistently observed excess mortality of Black/AA infants in the United States. There are a few relevant findings from this study. First, regardless of race, vaginal delivery was associated with decreased risk of infant mortality relative to cesarean section. Secondly, there were racial differences in the association between types of labor and delivery in infant mortality. Thirdly, irrespective of race, infant mortality was highest among mothers with less than high school education and lowest among mothers with a post-graduate degree. Fourthly, notwithstanding the method of delivery or race, the risk of dying as an infant was higher for males compared to females. Fifthly, there was a negative trend in period percent change, implying lower rates of infant mortality during the later period of the study (2012–2016).

The current study has demonstrated excess infant mortality of Black/AA relative to Whites. The observed racial differences in infant mortality had been illustrated by several studies to be three times as likely relative to Whites in Wisconsin [13]. Previous studies identified low socioeconomic status, maternal stressors, maternal education, and psychosocial stressors as exposure functions in infant mortality [14,15,16]. This disproportionate burden of infant mortality by race implies a survival disadvantage of Black/AA infants, and maybe explained by the social gradient in health including (though not limited to) the social determinates of health. Additionally, domestic violence during pregnancy as well as pre-conception stressors associated with family structure and violence have been implicated in infant mortality risk [12,17]. The current study intended to explain the observed racial differences in infant mortality by the method of delivery, namely vaginal and C-section. Specifically, the excess mortality of Black/AA infants relative to Whites appears to be explained by the marginalized differences between the vaginal and C-section method of deliveries in Black/AA. There are no studies to our knowledge that have assessed racial differences in infant mortality utilizing these exposures as an explanatory model. Available clinical data have implicated C-section in an increased risk of infant mortality [18].

We have also demonstrated that there were racial differences in the association between types of labor and delivery in infant mortality. There are no data to support or negate our findings. In the current study, regardless of the race of infant, vaginal delivery was protective compared to C-section, while the difference in vaginal vs. C-section rates was higher in Whites, and the difference was very marginal among Black/AA. The racial variances in the rate differences comparing C-section to vaginal may very well explain the excess mortality of Black/AA infants relative to Whites. However, we are unable to explain the marginal differences between vaginal and C-section rates among Blacks/AA in infant mortality. Notwithstanding, it is plausible to suggest that Black mothers may be provided with incompetent care at labor and delivery, suggestive of intrapartum clinicians’ believability, implying that Black mothers are used to pain and therefore ignored by the healthcare providers [19]. This racial bias in pain perception results in clinician delays in response to patients’ complaints, hence poor prognosis [19]. The implicated maternal stress at labor may induce pathologic changes in the neonates, resulting in excess infant mortality among Black/AA.

Our data have clearly indicated that regardless of race, infant mortality was highest among mothers with less than high school education and lowest among mothers with a post-graduate degree. Previous studies have indicated an inverse correlation between maternal education and infant mortality implying that the less educated the mother, the higher the infant mortality rate [20]. However, some studies have indicated that among Black/AA regardless of the mothers’ education, infant mortality rate does not vary [21]. The observed inconsistency or perceived deviation from the exposure function of education in infant mortality risk reduction may be explained by work related stress and employment instability among Black mothers [22]. The implication of workplace stress serves as a confounding variable in the association between higher education and lower infant mortality, and may be explained by the gene and job environment interaction. The work environment as a human social condition defined by low socioeconomic status, psychosocial stressors, isolation, and unstable social class may predispose Black/AA mothers to increased elaboration of conserved transcriptional response to adversity gene (CTRA). Such elaboration has been implicated in increased pro-inflammatory cytokine, such as interleukin-6, and decreased synthesis of immunoglobulin G (IgG), such as (antibodies synthesis), as well as decreased interferon gamma production, as an innate response to viral pathogens [23,24]. The overall molecular level events in this context may result in the constriction of the uterine blood vessels inducing fetal hypo-perfusion and the subsequent pathologic outcomes in the neonate increasing infant mortality [23].

This study has also illustrated that irrespective of the method of delivery or race, the risk of dying as an infant was higher for males compared to females. The observed sex variance in infant mortality has been previously observed [25]. Whereas genetic stability based on the sex chromosome, mainly XX and XY, have been utilized to explain the survival advantage of human female species. The epigenomic stability might provide additional information on such observations. The constantly observed survival advantage of female infants is explained by the interaction between the gene in the sex chromosomes and endogenous environment, such as hormones, as well as exogenous environment, such as maternal exposure to hypomethylated diet during pregnancy [26]. Epigenomic modulation involves the DNA methylation process as well as histone protein acetylation as a post transcriptional activity. The application of these two mechanistic processes may explain the transcriptional factors of deregulation and the subsequent impaired gene expression, while the histone acetylation process may restrict the availability of the DNA and the subsequent gene expression required for cellular, tissue, organ, and system functionality, enhancing infant mortality [27,28]. Because epigenomic modulation resulting in aberration is transgenerational (but reversible, especially at gametogenesis and in-utero), the adverse consequence for the fetus, neonate, and infant remains remarkable in influencing survival [29].

There was a negative trend in period percent change, implying lower rates of infant mortality during the later period of the study (2012–2016). These findings indicate a negative temporal trend in infant mortality rate. Whereas infant mortality has been shown to decrease with time, as confirmed by the current data, there remains racial variances or gaps in infant mortality. Despite the observed infant mortality trend reduction, the United States remains unfavorable in comparison with other industrialized nations in the world with respect to the rankings [16]. The observed low ranking of the U.S. as the 45th globally with respect to infant mortality is due in major parts by racial differences in the rates, implying health equity transformation. This approach involves resource allocation, proportionate universalism, implicit bias, and marginalization, where Blacks/AA are socially disadvantaged from birth, requiring equitable allocation of resources in growth, development, and early education, as well as the healthcare system treatment of Blacks/AA mothers and infants with equitable values in patient care.

Overall the observed racial differences in infant mortality is driven by a structural and organized social system that restricts the opportunity for the Black/AA subpopulation in the U.S. to benefit from social, economic, and environmental conditions related to human health. Specifically, improving the living conditions, increasing access to care, and care utilization (such as prenatal visits and equitable intrapartum care during delivery and labor), as well as minimizing clinician bias may reduce the excess infant mortality of Black/AA infants in the U.S. Furthermore, providing the opportunity in which living conditions are improved among Black mothers will offer the opportunity to improve the health of Black/AA children, thus marginalizing the Black–White risk difference in infant mortality. In addition, because the social gradient reflects environmental neighborhood characteristics, the understanding of gene and environment interaction, such as living conditions, may provide an additional strategic approach in intervention mapping. In effect, examining the gene and environment interaction, observed as epigenomic studies, will provide substantial data on interventions in narrowing the gaps between Blacks and Whites with respect to infant mortality. Epigenomic changes, which commences at gametogenesis, is transgenerational but reversible. The social signal transduction that is evoked from the stress placed on Blacks/AA mothers has a substantial effect on sympathetic nervous system and provokes the beta-adrenergic receptors. This response has been shown to provoke the conserved transcriptional response to adversity (CTRA) gene expression and the consequent elaboration of pro-inflammatory cytokine due to the impaired gene expression of the transcription factors and the inhibition of gene expression with respect to anti-inflammatory response [30]. In understanding these pathways of genomic stability and their role in disease causation as well as mortality, epigenomic studies are necessary in determining whether or not Black mothers, relative to White mothers have an increased mean deoxyribonucleic acid (DNA) methylation index with respect to the genome-wide analysis. Such initiatives will involve the utilization of the bisulfite pyrosequencing, which is very specific in differentiating between the methyl group and hydroxyethyl group, as well as the binding of these groups to the Cytosine-phosphate-Guanine (CpG) region of the gene, inhibiting transcription and the messenger ribonucleic acid (mRNA) sequencing, leading to impaired gene expression and abnormal cellular functionality. The reference to epigenomic investigations reflects the inability of an infant to respond to treatment modalities due to the drug receptors in-availability, resulting from impaired gene expression (mRNA translation dysregulation) [31]. The observed epigenomic aberration clearly illustrates treatment effect heterogeneity, in which some subpopulations respond differentially to a given therapeutic agent in the phase of epigenomic lesion, explaining infant mortality racial risk differences.

Despite the large sample size utilized in this study and the rigorous methodology that is very novel in aggregate data modelling, there are some limitations. First, as a secondary data analysis, there remains a potential for information and misclassification biases. Secondly, due to the nature of the data and design (i.e., cross-sectional design), there remains a tendency for reverse causation, which is unlikely given mortality as the response variable with the method of delivery as the main explanatory variable in this modeling. Thirdly, these findings may be driven by unmeasured confounding variables, such as maternal socioeconomic status (SES). However, it is highly unlikely that the implication of the method of delivery in explaining the excess infant mortality among Black/AA is driven solely by these unmeasured confounding variables [32]. Other limitations of this study include the potential for misclassification of infant race.

## 5. Conclusions

In summary, there are racial disparities in infant mortality, which were explained in part by the labor and delivery method comparing vaginal to cesarean section. However, due to the aggregate nature of these data, caution is required in the application of these findings and in intervention mapping in reducing the Black–White infant mortality risk differences. Further, these findings recommend effective policy formulation, implementation, and evaluation in understanding the cause of causes mainly, social gradient in addressing the perpetually and persistently observed racial gap in infant mortality in the USA.

## Figures and Tables

**Figure 1 ijerph-17-03146-f001:**
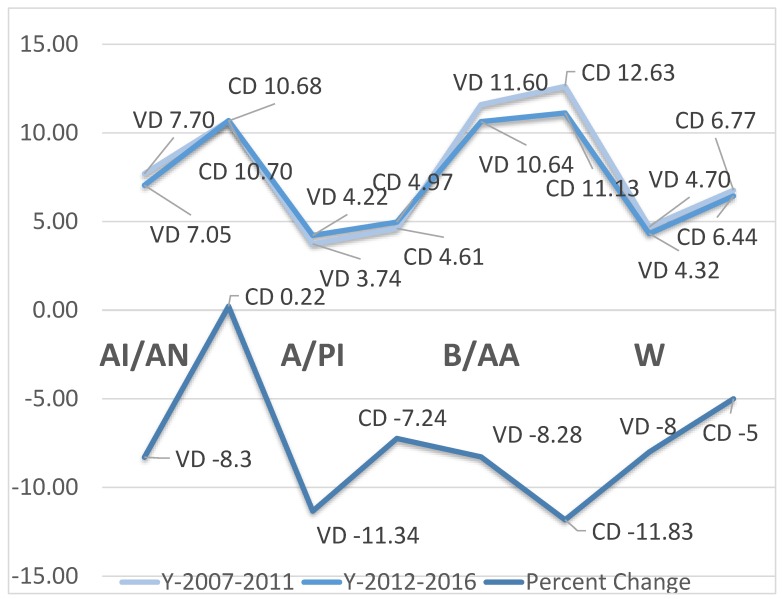
Trends and period percent changes. AI/AN, American Indian/Alaskan Native; A/PI, Asian/Pacific Islander; B/AA, Black/African American; CD, cesarean delivery; VD, vaginal delivery; W, White; Y, year.

**Table 1 ijerph-17-03146-t001:** Cumulative incidence of infant mortality by method of delivery stratified by race, 2007–2016.

Variable	AI/AN *n* (%)	A/PI *n* (%)	B/AA *n* (%)	W *n* (%)	$x^{2}$ (df)	*p*-Value
Method of Delivery					40,000 (4)	<0.001
Vaginal	2459 (0.7)	7073 (0.4)	46,709 (1.1)	94,737 (0.5)		
Cesarean	1408 (1.1)	4179 (0.5)	26,960 (1.2)	64,890 (0.7)		

Notes: AI/AN, American Indian/Alaskan Native; A/PI, Asian/Pacific Islander; B/AA, Black/African American; W, White.

**Table 2 ijerph-17-03146-t002:** Incidence rate ratio in infant mortality by race 2007–2016.

Variable	Incidence Rate Ratio	95% CI	*p*-Value
Race			
AI/AN	0.69	0.64–0.74	<0.001
A/PI	0.83	0.80–0.86	<0.001
B/AA	0.94	0.92–0.95	<0.001
W	0.68	0.67–0.69	<0.001

Notes: AI/AN, American Indian/Alaskan Native; A/PI, Asian/Pacific Islander; B/AA, Black/African American; CI, Confidence Interval; W, White.

**Table 3 ijerph-17-03146-t003:** Infant mortality rate by mothers’ education and method of labor and delivery.

Method of Delivery	Maternal Education	Mother Race	Infant Mortality Rate
Vaginal	<High school	AI/AN	9.33
Cesarean	<High school	AI/AN	14.3
Vaginal	Bachelor’s degree	AI/AN	3.53
Cesarean	Bachelor’s degree	AI/AN	5.88
Vaginal	<High school	A/PI	4.95
Cesarean	<High school	A/PI	7.58
Vaginal	Post-graduate degree	A/PI	2.63
Cesarean	Post- graduate degree	A/PI	3.29
Vaginal	<High school	B/AA	9.8
Cesarean	<High school	B/AA	13
Vaginal	Post-graduate degree	B/AA	7.46
Cesarean	Post-graduate degree	B/AA	6.45
Vaginal	<High school	W	5.5
Cesarean	<High school	W	8.85
Vaginal	Post-graduate degree	W	2.5
Cesarean	Post- graduate degree	W	3.9

Notes: AI/AN, American Indian/Alaskan Native; A/PI, Asian/Pacific Islander; B/AA, Black/African American; CI, Confidence Interval; W, White.

**Table 4 ijerph-17-03146-t004:** Incidence rate ratio in infant mortality by race and gender, 2007–2016.

Variable	Method of Delivery	Gender	Incidence Rate Ratio	95% CI	*p*-Value
Race					
AI/AN	Vaginal	Male			
		Female	0.8	0.74–0.87	<0.001
	Cesarean	Male			
		Female	0.82	0.74–0.91	<0.001
A/PI	Vaginal	Male			
		Female	0.78	0.74–0.81	<0.001
	Cesarean	Male			
		Female	0.91	0.85–0.97	<0.001
B/A	Vaginal	Male			
		Female	0.79	0.78–0.81	<0.001
	Cesarean	Male			
		Female	0.89	0.87–0.91	<0.001
W	Vaginal	Male			
		Female	0.81	0.79–0.81	<0.001
	Cesarean	Male			
		Female	0.89	0.88–0.91	<0.001

Notes: AI/AN, American Indian/Alaskan Native; A/PI, Asian/Pacific Islander; B/AA, Black/African American; CI, Confidence Interval; W, White.

**Table 5 ijerph-17-03146-t005:** Cumulative infant mortality rates by five-year intervals and the percent change by race and method of labor and delivery.

Race	Method	2007–2011	2012–2016	Percent Change
AI/AN	Vaginal	7.70	7.05	−8.3
	Cesarean	10.68	10.70	0.22
A/PI	Vaginal	3.74	4.22	−11.34
	Cesarean	4.61	4.97	−7.24
B/AA	Vaginal	11.60	10.64	−8.28
	Cesarean	12.63	11.13	−11.83
W	Vaginal	4.70	4.32	−8
	Cesarean	6.77	6.44	−5

Notes: AI/AN, American Indian/Alaskan Native; A/PI, Asian/Pacific Islander; B/AA, Black/African American; W, White.

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
