# Peer review of "Implication of Vaginal and Cesarean Section Delivery Method in Black–White Differentials in Infant Mortality in the United States: Linked Birth/Infant Death Records, 2007–2016"

_ijerph, 2020, doi:10.3390/ijerph17093146_

Round 1

Reviewer 1 Report

I have read this paper with great interest, and i have some specific comments in addition to textual suggestions. 

This is an association, not for sure reflecting causality, highly recommend to rephrase the title.

What do we know about the caesarean, is this related to preterm delivery, or emergency/secondary caesarean during labor ? and is the infant mortality mainly perinatal or post neonatal setting ?  

I fully respect and regret the data as provided, but the lines 40-41 are likely too simplistic, what about level of teaching, socio-economic, smoking, breastfeeding ?

How does the 6.16/1000 cumulative infant mortality compared other ‘first world countries’

The epigenomic part of the discussion is an interesting hypothesis, but the discussion of the study to be done ? is obsolete in my reading,.

specific comments

Line 15: increase in mortality = reads perhaps better as increase in infant mortality.

Line 16: eclampsia, or preeclampsia

Line 17: SES ? abbreviation ? and cause of causes ?

Line 67: what do the authors mean with ‘even though vaginal deliveries already double’ ? is vaginal delivery not the reference type of delivery ?

Line 72: 31.9 to 32 % is not a real increase, suggest to rephrase or to describe the 2007 to 2016 trend, if possible (ref NCHS 2007-2016).

Author Response

Dear Editor,

Thank you and the reviewers for the opportunity to review our paper for resubmission for consideration for publication by your journal. We have carefully examined the reviewers’ comment and observed them to be valuable in contributing to the knowledge of infant mortality racial differences in the US. Below please see the authors’ responses:

Reviewer’s Comment: I have read this paper with great interest, and i have some specific comments in addition to textual suggestions.

Authors’ Response: Thanks for the observations, suggestions and comments which had been very relevant in enhancing the readability of the paper as well as addressing some methodologic issues in terms of causal inference .

Reviewer’s Comment : This is an association, not for sure reflecting causality, highly recommend to rephrase the title.

Authors’ Response: Thanks for the observation. As an author of “Applied epidemiologic principles and concepts” , I very firmly accept your suggestion. This had been carefully addressed. https://www.routledge.com/Applied-Epidemiologic-Principles-and-Concepts-Clinicians-Guide-to-Study/Jr/p/book/9781498733786

Reviewer’s Comment : What do we know about the caesarean, is this related to preterm delivery, or emergency/secondary caesarean during labor ? and is the infant mortality mainly perinatal or post neonatal setting ?

Authors’ Response: Thanks for the comment. C-section had been explained and it embraces planned /scheduled and emergency C-section. Also infant mortality is characterized in the paper to begin from neonatal, thus not including some perinatal mortality since perinatal mortality also includes neonatal.

Reviewer’s Comment : I fully respect and regret the data as provided, but the lines 40-41 are likely too simplistic, what about level of teaching, socio-economic, smoking, breastfeeding ?

Authors’ Response: Thanks for your consideration of other predisposing factors to IM . We have addressed this comment by describing exclusive breast feeding and low SES which was implicated in the discussion in explain the race differentials in IM.

Reviewer’s Comment :How does the 6.16/1000 cumulative infant mortality compared other ‘first world countries’

Authors’ Response: Thanks! We have addressed this by implicating US and ranked 45th globally with respect to IM.

Reviewer’s Comment : The epigenomic part of the discussion is an interesting hypothesis, but the discussion of the study to be done ? is obsolete in my reading,.

Authors’ Response: Thanks for this comment. In addressing risk factors and addressing for potential confounding, it is relevant to consider other factors if we assume that we have utilized if feasible all confounding in the model, and yet variances still remain. Available epidemiologic data implicate the NR3C1 gene in psychosocial stressors associated with preterm birth. Since preterm birth has a significant role in IM, it was our scientific commitment to explain and alternative approach by explaining the overall role of gene and environment differentials in driving pathology is some population relative to others. Allowing this section in the paper, which had been simplified, thanks to your comment, will encourage studies in this direction. The primary author in this work is a translational epigenomist and is willing to simplify further this aspect of the discussion, should we perceive this material t be too technical for public health consumption.

Reviewer’s Comment : specific comments

Reviewer’s Comment Line 15: increase in mortality = reads perhaps better as increase in infant mortality.

Authors’ Response: Thanks, addressed, please see the bold, and remove the bold (unbold) prior to publication

Reviewer’s Comment :Line 16: eclampsia, or preeclampsia

Authors’ Response: Thanks, addressed please see the bold, and unbold prior to publication,

Reviewer’s Comment :Line 17: SES ? abbreviation ? and cause of causes ?

Authors’ Response: Thanks, addressed please see the bold, and unbold prior to publication

Reviewer’s Comment : Line 67: what do the authors mean with ‘even though vaginal deliveries already double’ ? is vaginal delivery not the reference type of delivery ?

Authors’ Response: Thanks, addressed, please see the bold, and unbold prior to publication.

Reviewer’s Comment : Line 72: 31.9 to 32 % is not a real increase, suggest to rephrase or to describe the 2007 to 2016 trend, if possible (ref NCHS 2007-2016).

Reviewer 2 Report

IJERPH

The first sentence of the introduction is written in a confusing way and should be revised.

Lines 88-89: Should be “aimed”; also, the rest of the sentence is confusing. Be more explicit and direct in the study aims.

Lines 94-95: run-on sentence. Please revise.

Line 94: be more explicit about IRB approval. Which institution approved the study?

Line 99: United State should be United States

Line 106-111: provide a reference for this “cross-sectional ecologic design”… Also, if this study is using an ecologic design, then why not include more ecological measures? Mode of delivery does not necessarily tap into ecologic factors. Neither does education.

Line 109: what is meant by “and are accurate…”?

Line 121: remove the comma after “difference”

Line 126: remove the comma after “while”

Line 135: “test” should be “tests”

Line 139: remove the period after deaths

Analyses: Explicitly state the statistics used to analyze the data and whether or not the outcome variables were normally distributed. Should consider using a mixed effects model given the race/mode of delivery combinations

Results: Results in Table 3 are meaningless without more explicit statistical tests indicating differences across education.

Line 148: “there were” – not “there was racial differences”

Line 253: “stressors”

Line 272: “maybe” should be two words

Line 274: “ignorability” is not a word

The discussion spends too much time discussing epigenetics without using any references. First of all, this lengthy explanation has nothing to do with the purpose of the paper, but the words are closely matched to other sources (not to mention the authors do not cite the information they are discussing). Then, the authors state that infant mortality is driven by structural factors, yet the whole section on epigenetics has nothing to do with structural factors  - that is genetics. Furthermore, mode of delivery and educational factors alone say nothing about structure or social systems.  

Limitations: to what “confoundings” are the authors referring?

Author Response

Dear Editor,

Thank you and the reviewers for the opportunity to review our paper for resubmission for consideration for publication by your journal. We have carefully examined the reviewers’ comment and observed  them  to be valuable  in contributing to the knowledge of infant mortality racial differences in the US. Below please see the authors’ responses:

Reviewer’s  Comment : The first sentence of the introduction is written in a confusing way and should be revised.

Authors’  Response:  Thanks for the comment, which had been  addressed, please send the bold.

Reviewer’s  Comment Lines 88-89: Should be “aimed”; also, the rest of the sentence is confusing. Be more explicit and direct in the study aims.

Authors’ Response: Thanks for the comment, which had been  addressed, please send the bold.

Reviewer’s  Comment Lines 94-95: run-on sentence. Please revise.

 Authors’ Response:

Reviewer’s  Comment Line 94: be more explicit about IRB approval. Which institution approved the study?

 Authors’ Response: Thanks for the comment, which had been  addressed, please send the bold.

Reviewer’s  Comment Line 99: United State should be United States

 Authors’ Response: Thanks for the comment, which had been  addressed, please send the bold.

Reviewer’s  Comment Line 106-111: provide a reference for this “cross-sectional ecologic design”… Also, if this study is using an ecologic design, then why not include more ecological measures? Mode of delivery does not necessarily tap into ecologic factors. Neither does education.

 Authors’ Response: Thanks for the comment, which had been  addressed, please send the bold.

Reviewer’s  Comment : Line 109: what is meant by “and are accurate…”?

 Authors’ Response: Thanks for the comment, which had been  addressed, please send the bold.

Reviewer’s  Comment Line 121: remove the comma after “difference”

 Authors’ Response: Thanks for the comment, which had been  addressed, please send the bold.

Reviewer’s Comment Line 126: remove the comma after “while”

 Authors’ Response: Thanks for the comment, which had been  addressed, please send the bold.

Reviewer’s  Comment Line 135: “test” should be “tests”

 Authors’ Response: Thanks for the comment, which had been  addressed, please send the bold.

Reviewer’s  Comment Line 139: remove the period after deaths

 Authors’ Response: Thanks for the comment, which had been  addressed, please send the bold.

Reviewer’s  Comment Analyses: Explicitly state the statistics used to analyze the data and whether or not the outcome variables were normally distributed. Should consider using a mixed effects model given the race/mode of delivery combinations

 Authors’ Response: Thanks for the comment, which had been  addressed, please send the bold.

Reviewer’s  Comment Results: Results in Table 3 are meaningless without more explicit statistical tests indicating differences across education.

Authors’ Response: Thanks for the observation. Unfortunately we do not agree with the comment since descriptive  statististics , which is non-inferential does not require an inferential test. The primary author of this paper is a professor of clinical stsiustuitics ad had published in stats modeling: https://www.routledge.com/Applied-Biostatistical-Principles-and-Concepts-Clinicians-Guide-to-Data/Holmes-Jr/p/book/9781498741194

Reviewer’s  Comment Line 148: “there were” – not “there was racial differences”

 Authors’ Response: Thanks for the comment, which had been  addressed, please send the bold.

Reviewer’s  Comment Line 253: “stressors”

 Authors’ Response: Thanks for the comment, which had been  addressed, please send the bold.

Reviewer’s  Comment Line 272: “maybe” should be two words

 Authors’ Response: Thanks for the comment, which had been  addressed, please send the bold.

Reviewer’s  Comment Line 274: “ignorability” is not a word

 Authors’ Response: Thanks for the comment, which had been  addressed, please send the bold.

Reviewer’s  Comment The discussion spends too much time discussing epigenetics without using any references. First of all, this lengthy explanation has nothing to do with the purpose of the paper, but the words are closely matched to other sources (not to mention the authors do not cite the information they are discussing). Then, the authors state that infant mortality is driven by structural factors, yet the whole section on epigenetics has nothing to do with structural factors  - that is genetics. Furthermore, mode of delivery and educational factors alone say nothing about structure or social systems.  

Authors’ Response: Thanks for this comment. In addressing risk factors and addressing for potential confounding, it is relevant to consider other factors if we assume that we have utilized if feasible all confounding in the model, and yet variances still remain. Available epidemiologic data implicate the NR3C1 gene in psychosocial stressors   associated with preterm birth.  Since preterm birth has a significant role in IM, it was our scientific commitment to explain and alternative approach by explaining the overall role of gene and environment differentials in driving pathology is some population relative to others.  Allowing this section in the paper, which had been simplified, thanks to your comment, will encourage studies in this direction.  The primary author of  this work is a translational epigenomist and  is willing to simplify further this aspect of the discussion, should we perceive this material to be too technical for public health consumption.

Reviewer’s  Comment Limitations: to what “confoundings” are the authors referring?

Autors’ Response: Authors’ Response: Thanks for the comment, which had been  addressed, please send the bold.

Round 2

Reviewer 1 Report

no additional comments 

Author Response

This study utilized a public accessed aggregate data from the Center for Disease control (CDC) hence there was no need for ethics code.

Reviewer 2 Report

Several issues still exist with typos:

Ln 77 - "tow" should be "two"

Ln 98 - factor should be factors

Ln 109 - these data set?

Ln 266 - white should be whites

Ln 267 - stress should be stressors

Ln 276 - accessed? maybe assessed instead?

Ln 291 - need a period

Ln 305-308 - need a reference

Ln 314-324 - needs a reference

Ln 333 - grammar issues; does not really follow logically from prior statement

Ln 354 - need a reference

Ln 370 - no comma needed

Ln 371 - "as pre-existing data" - does not make sense.

Author Response

(The authors gave the same response as above.)
